# Cryptococcosis: Identification of Risk Areas in the Brazilian Amazon

**DOI:** 10.3390/microorganisms10071411

**Published:** 2022-07-13

**Authors:** Danielle Saraiva Tuma dos Reis, Mioni Thieli Figueiredo Magalhães de Brito, Ricardo José de Paula Souza Guimarães, Juarez Antônio Simões Quaresma

**Affiliations:** 1Faculty of Nursing, Federal University of Pará (UFPA), Belém 66075-110, PA, Brazil; 2Faculty of Pharmacy, Federal University of Pará (UFPA), Belém 66075-110, PA, Brazil; mionibrito@gmail.com; 3Geoprocessing Laboratory, Evandro Chagas Institute (IEC), Ananindeua 67030-000, PA, Brazil; ricardoguimaraes@iec.gov.br; 4Juarez Antônio Simões Quaresma Tropical Medicine Center, Federal University of Pará (UFPA), Belém 66075-110, PA, Brazil; juarez.quaresma@gmail.com

**Keywords:** cryptococcosis, time-series studies, spatial analysis, epidemiological profiles

## Abstract

The Brazilian Amazon has a specific epidemiological profile for cryptococcosis, considering its social and economic inequality, health reality, and low access to health services. Furthermore, Brazil and Colombia have the highest cryptococcosis incidence rates in Latin America. In this study, we identified the areas of risk for cryptococcosis in the state of Pará in the Brazilian Amazon. This was an ecological study of patients admitted to a referral hospital from 2008 to 2018, aged 13 years or older, and of both sexes. The spatial distribution was determined using ArcGis 10.3.1 software. Cryptococcosis was confirmed in 272 cases. The incidence rate was 3.41 cases/100,000 inhabitants. Spatial distribution was concentrated in the Metropolitana de Belém, Nordeste Paraense, and Marajó mesoregions. The sociodemographic profile consisted of 62% men, aged between 24 and 34 years (36%), without completed secondary education (64.7%), and with occupations varying between agricultural activities (13.8%) and household activities (22%). The mean hospitalization time was 39 days; the prevalent clinical form was neurological (89.7%). The mortality rate among patients with cryptococcosis was up to 40%. Knowledge of the real magnitude of the disease in the Brazilian Amazon makes it possible to identify areas with the greatest risks and to propose control and epidemiological surveillance programs.

## 1. Introduction

Brazil is one of the countries with the highest inequality rates according to the 2016 Global Human Development Report of the United Nations Development Program (UNDP), ranking 10th out of 143 countries. Recognizing and combating this inequality is a complex and permanent challenge for Brazilian society [1]. Pará, a state in Brazil located in the North Region, is part of the Brazilian Amazon and is the second largest area of federative units, with an estimated population of 8 million inhabitants and alarming health characteristics [2].

This reflects illnesses resulting from precarious living conditions in the Brazilian Amazon and low access to health services, evidencing an epidemiological profile marked by infectious and parasitic diseases. Recent economic growth did not correspond to social development, as expressed through indicators such as the Human Development Index/HDI, which in Pará is equivalent to 0.646 below the average Brazilian standard, a value characterizing neglected populations [2].

Consequently, health demands fall on establishments such as the João de Barros Barreto University Hospital (JBBUH), a reference in the state capital for the care of patients with infectious and parasitic diseases, including cryptococcosis [3].

Cryptococcosis is an emerging human fungal infection that presents mainly in the form of meningoencephalitis [4]. Until the 1960s, it was considered a rare disease with little clinical importance. In the 1980s, with the emergence of the human immunodeficiency virus (HIV), there was an increase in the number of infections caused by Cryptococcus, making it one of the most prevalent systemic infections worldwide [5,6,7].

In an estimate of the global incidence of the disease based on reports by the Joint United Nations Program on HIV-AIDS (UNAIDS) and the World Health Organization (WHO), Rajasingham et al. [8] predicted that cryptococcal meningitis would result in 15% AIDS-related mortality, confirming that the disease is the second most common cause of mortality in these patients, surpassed only by tuberculosis. It is believed to affect more than a million people annually worldwide [9].

In Latin America, this mycosis affects over 5000 people with cryptococcal meningitis each year, and there is a total of 2400 deaths [10]. In Brazil, surveillance is based on case series analysis, diagnosed in some regional centers, or indirect data obtained from the AIDS program. A Brazilian mortality study identified 5755 deaths, of which 4634 (80.5%) were associated with AIDS and other immunosuppressive diseases, and 1121 (19.5%) were the underlying causes of death [11]. A different scenario was observed in developed countries, especially Canada, where there is a control and epidemiological surveillance program for the disease, with a fatality rate of 8.7% [12].

We investigated Pará to identify risk areas to obtain information on temporal trends of incidence under the sociodemographic and clinical–laboratory aspects to determine the magnitude of this disease in the Brazilian Amazon. These estimates will serve to propose control and epidemiological surveillance programs, such as screening programs for cryptococcal antigens in patients with HIV/AIDS.

## 2. Materials and Methods

An observational, descriptive–analytical, ecological-type study of a historical series with a spatiotemporal analysis of cryptococcosis cases [13,14,15] was conducted at JBBUH, a university hospital located in Belém, the capital of Pará, where medium- and high-complexity care is offered by the Unified Health System (SUS). It covers 26,420 m^2^, offering 212 infirmary beds and 6 ICU beds and is registered with the Ministry of Health, a regional reference for communicable diseases [3].

The study data involved medical records of patients hospitalized at the JBBUH, from January 2008 to December 2018, aged 13 years or older, of both sexes, residing in Pará, with a confirmed diagnosis of cryptococcosis by direct examination of the cerebrospinal fluid (CSF), histopathological evidence at any site, and/or confirmation of fungal growth in culture media.

Sociodemographic and clinical laboratory data were collected and recorded in an Excel 7.0 database (DB), with frequency distribution and average calculations. The Biostat 5.3 program was used for statistical analysis, with a significance level of 5%, for the G tests and the chi-squared test with residue analysis evaluation according to each test’s assumptions.

The incidence rate was calculated as the number of patients with cryptococcosis divided by the population of the municipalities of origin, multiplied by 100,000 inhabitants. According to the Brazilian Institute of Geography and Statistics (IBGE) projections, the population was obtained based on the 2010 demographic census and recent birth and death records [2].

The cases were spatialized according to the mesoregions of the state. Pará is composed of six mesoregions: Metropolitana de Belém (MB), Northeast Pará (NP), Marajó (MJ), Baixo Amazonas (BA), Southeast Pará (SPa), and Southwest Pará. The geographic database (BDGeo) was created from the interrelationship of epidemiological, clinical, demographic, and cartographic databases indexed by geographic coordinates obtained through the LAT/LONG Projection System with DATUM SIRGAS 2000.

The spatial distribution of cryptococcosis cases was determined using ArcGIS 10.3.1 and classified according to incidence rates (absence, low, medium, high, and very high) through choropleth maps according to coloring based on the distribution of quartiles [16].

The study was approved by the JBBUH ethics committee (CAAE number: 62013716.1.0000.0017). Anonymity, privacy, image protection, non-stigmatization, and non-use of information to the detriment of the participants were maintained following resolution 466 (2012) of the National Health Council [17].

## 3. Results

Approximately 272 cases of cryptococcosis were confirmed from 2008 to 2018, of which 170 (62.5%) were HIV co-infections. The mortality rate was 40.1% (N = 109). The sociodemographic profile corresponded to the majority of men (62%) aged between 24 and 34 years (36%) and between 35 and 45 years (28.3%). Approximately 64.7% of the patients did not complete high school. Professional occupations varied among household activities (22%), agricultural activities (13.8%), and students (13%), according to Table 1.

The clinical and laboratory characteristics are shown in Table 2, with an average hospitalization time of 39 days (range, 228 days). The time of disease onset was in the subacute form (46%), with clinical manifestations of the headache triad (90.1%), fever (76.1%), and vomiting (71.3%). For the diagnosis of the disease, in most cases, the hospital carried out direct detection of fungus in CSF with India ink staining (98.2%), CSF characteristics representing high cellularity (64.7%), with a predominance of mononuclear cells (88.9%), glycorrhachia (55.1%), and proteinorrhachia (48.7%). Although 70.9% had performed fungal culture, it was not possible to isolate the cryptococcal species (*C. neoformans* or gattii species complex), and 10.7% showed the formation of cryptococcoma, both in pulmonary and neurological involvement through tomography.

Meningoencephalitis was the most common form of cryptococcosis infection (89.7%), followed by cases associated with the pulmonary form (6.6%), and the third most frequent were deep fungal infection causing fungemia (1.8%) and cutaneous forms (1.1%).

Regarding the disease sequelae, 30 (18.4%) cases reported visual deficits, followed by hearing deficits (4.3%), limb paresis and paralysis (3.7%), and pulmonary sequelae (2.5%) (pulmonary cryptococcoma, pulmonary fibrosis and lobectomy).

Table 3 shows the clinical evolution of 215 patients who received antifungal treatment during hospitalization. For combined treatment, the majority of patients showed clinical improvement (78.7%). However, for those who received only amphotericin B deoxycholate (AmB), 38 (60.3%) died in the initial induction phase before fluconazole was started (consolidation phase).

Figure 1 shows the distribution of new cases of cryptococcosis over the years, with an increase, an average of 25 hospitalizations per year, and an incidence rate of 3.41 cases per 100,000 population (272/7,969,654) over 11 years.

Table 4 presents the clinical evolution of patients according to their present immunological condition, which had a mortality rate 47.1% higher in patients co-infected with HIV.

Results of the analysis of incidence rates by region, indicate that for the mesoregions, the occurrence of cryptococcosis was higher in the Metropolitana de Belém (MB), with 6.7 cases per 100,000 inhabitants, followed by Northeast Pará (NP) with 4.07 cases per 100,000 inhabitants, Marajó (MJ) with 1.52 cases per 100,000 inhabitants, Southeast Pará (SPa) with 0.98 cases per 100,000 inhabitants, and Baixo Amazonas (BA) with 0.13 cases per 100,000 inhabitants.

In the choroplethic map presented in Figure 2, the concentration of cryptococcosis cases is shown for MB, MJ, and NP (quartile 6.63–18.52), which are neighboring mesoregions geographically located on the Brazilian coast and closer to the state capital where the referral hospital is based. In addition, the dimension of the state in relation to the 144 municipalities can be observed by the spatialization of cases. There is a lack of registration in Southwest Paraense and few municipalities identified cases in BA and SPa, far from the capital.

## 4. Discussion

The epidemiological panorama of cryptococcosis in Pará corroborates the findings in Brazilian states from the north to south [18,19,20], with a prevalent profile in men, young adults, those having little education, and being associated with immunosuppressive diseases (HIV/AIDS), and manifesting itself in a subacute and meningoencephalitis clinical form.

It is a disease with a long onset due to a lack of clinical suspicion, especially in immunocompetent patients, and a long hospitalization time. For this reason, the WHO [21] has recommended as a diagnostic alternative, the search for cryptococcal antigen (CrAg) in HIV-infected patients with low CD4 levels and neurologically asymptomatic patients. This method allows the identification of cryptococcal disease at a subclinical stage, on average, 3 weeks before the onset of symptoms of cryptococcal meningitis, as a means of diagnostic screening and subsequent treatment of patients [22].

For the Amazonian reality, this test has the potential to be used in locations where logistics and techniques preclude the use of other tests, do not require refrigeration or advanced laboratory equipment, and can be stored at room temperature for up to 1 year [23].

Regarding treatment, Aguiar et al. [20], in an epidemiological study similar to this study, found susceptibility to the tested antifungal agents (fluconazole, voriconazole, amphotericin B, and 5-flucytosine) in 100% of patients despite a mortality rate of 58.5%.

For the study participants, amphotericin B (AmB) and fluconazole were the treatments of choice, with a clinical improvement of 78.7%; however, this combination was compromised due to adverse events and the need for hospitalization for intravenous administration and monitoring of electrolytes [24,25,26]. Fluconazole monotherapy is not recommended because of higher mortality, unfavorable clinical outcomes, and the risk of drug resistance [27]. AmB lipid formulations have been associated with a lower risk of nephrotoxicity; however, there is still controversy about the differences between the two available lipid formulations [26].

Cryptococcal meningitis is a serious opportunistic infection in immunocompromised individuals, and it is estimated that 223,100 cases of the disease result in 181,000 deaths annually among people living with HIV [21].

A study in Brazil on the total number of deaths in which cryptococcosis was mentioned in the statements registered in the Mortality Information System (SIM) from 2000 to 2012 reported 5755 deaths. Of these, 4634 (80.5%) deaths were associated with AIDS and other immunosuppressive diseases, and in 1121 (19.5%), it was the underlying cause of death. When verifying the distribution according to the federated units (UF) in Brazil, the north region had the highest mortality rates due to an underlying cause, with 0.63/million inhabitants. Another study also showed a significant growth trend in the North Region in the mortality rate related to cryptococcosis [28], confirming the results of the research in question.

When the magnitude of AIDS cases reported in the north region was verified, concern increased, with the detection rate being 44.2% higher than that in other regions in Brazil. Belém had 23.6 cases per 100,000 inhabitants in 2017, and Pará ranked 4th with 7.8 deaths/100,000 inhabitants [29], representing a population at risk for opportunistic infections caused by yeast-like fungi of the Cryptococcus species complex.

Regarding the incidence rate of cryptococcosis in Brazil, only estimates or data on meningitis due to other etiologies were found due to the lack of mandatory notification for cryptococcal disease. As disclosed by the Notifiable Diseases Information System (SINAN) [30], from 2010 to 2018, the incidence rate for meningitis due to other etiologies ranged from 0.34 to 0.41 confirmed cases per 100,000 inhabitants. These values were close to those found in the study in question.

In Latin America, Brazil and Colombia stand out as the countries with the highest incidence rates, between 1001 to 2500 cases, followed by Argentina and Mexico, with incidence rates of 501 to 1000 cases [8].

However, epidemiological studies in these countries are very limited, despite cryptococcosis being considered one of the most common mycoses, with a prevalence of 10% in Mexico and 19% in Venezuela, occupying third place after histoplasmosis and paracoccidioidomycosis, and with a prevalence of 20% in Argentina as the second most frequent deep fungal infection, and in Colombia, it stands out as a common opportunistic mycosis in patients with HIV/AIDS, with a prevalence of 76% [10].

Leimann and Koifman [31] also evaluated the epidemiology of cryptococcal meningitis in Rio Janeiro from 1994 to 2004 and found an average incidence of 0.45 per 100,000 inhabitants, with a range from 0.30 to 0.58, similar to the findings of the present study. In another estimate of fungal diseases made by Giacomazzi [32], based on the Unified Health System of the Department of Informatics (DATASUS) for the year 2011, a rate of 3.52 per 100,000 inhabitants for meningitis cryptococcal in Brazil was found.

The Amazon region is considered to have an epidemiological profile for cryptococcosis, including social and economic inequality, health reality, and low access to health services. Pará is part of this region that comprises eight states in the north of Brazil, the second in terms of territorial dimension in Brazil, distributed in an area of 1.2 million km^2^ and a population density of 6.07 inhabitants/km^2^, constituting the least populous geoeconomic region [33]. Structured from horizontal social relations, often anchored in various social conflicts, mainly driven by the dispute for land with the significant deforestation of the primary forest, depletion of natural and mineral resources, and the non-internalization of the wealth produced in vital public services, it has a great impact on the quality of life [34,35].

The process of socioeconomic configuration of the state in the six mesoregions was influenced by the beginning of Portuguese colonization in the Amazon and the production of rubber from the 20th century onwards, which impacted the state economy [36]. In addition, the construction of highways allowed the migratory flow from the coast to the interior to the detriment of the exploitation of forest resources [34].

Communities mainly occupy the mesoregions of Pará and villages in the Amazon forest, share areas for cattle grazing, and exploit wood, except for Metropolitana de Belém, which comprises urbanization areas around the city and the port of Belém, with an extensive trade in wood by-products for local use and export [33].

It is noteworthy that among the mesoregions of Pará, there was no record in the southwest of Pará, which is considered the largest territorial area (one-third of the state area, with 415.8 thousand km^2^) and has the lowest demographic density (1.2 inhabitants per km^2^). This may have been due to various causes, such as geographic and displacement factors, with the likely migration of these patients to other public services in the region or to neighboring states in Brazil such as Amazonas and Mato Grosso [2].

In contrast, we have the MB mesoregion, with the municipalities of Ananindeua and Belém, the capital of the state, concentrating around 2700 and 1360 people per km^2^ in their territories, respectively. MB and NP comprise the majority of the state population, with approximately 4 million people [34]. The incidence rate was higher in these patients.

The pioneering economic activities in these locations are based on farming and açaí extraction. The slash-and-burn agricultural system is dominant in the production of subsistence crops, mainly cassava, corn, rice, and cowpea. Characterized by municipalities with small areas and low population density, some villages had (as their first inhabitants) descendants of northeastern people and immigrants, who constituted the so-called pioneer fronts in the 1950s in the construction of roads and bridges for the flow of agricultural production [37,38,39].

Associated with these factors, Pará has the highest deforestation rates among the states in the northern region. In 2014 alone, the state had a deforested area of 1829 km^2^ [40]. Deforestation can increase the number of cases, causing profound environmental changes, rural depletion, and increasing urbanization [41]. According to studies in One Health [42], changes on a global scale cause an imbalance in the ecosystem and lead to the emergence of infectious and non-infectious diseases; however, an interdisciplinary approach is essential to understanding human interactions with animals and ecosystems.

Thus, the construction of a health surveillance system, guided by a data analysis model distributed across geographic space, has been increasingly valued in health management as it points to new subsidies for the planning and evaluation of actions based on the analysis of the spatial distribution of diseases, the location of health services, and environmental risks, among other factors [16].

## 5. Conclusions

This study made it possible to realize the magnitude of the occurrence of cryptococcosis in the Brazilian Amazon, with 272 cases and an incidence rate of 3.41 cases per 100,000 inhabitants. This record reflects the fragility in the diagnosis and treatment of patients, with a mortality rate of 40%, and being more expressive in patients living with HIV and AIDS (47.1%, *p* = 0.0036).

Thus, the present study aimed to alert the competent bodies (surveillance services, health departments, government agencies) for the need to include cryptococcosis in the National Compulsory Notification List, despite being a regional estimate. Knowledge of the actual magnitude of the disease in the Brazilian Amazon and the spatialization of cases will allow the identification of risk areas.

## Figures and Tables

**Figure 1 microorganisms-10-01411-f001:**
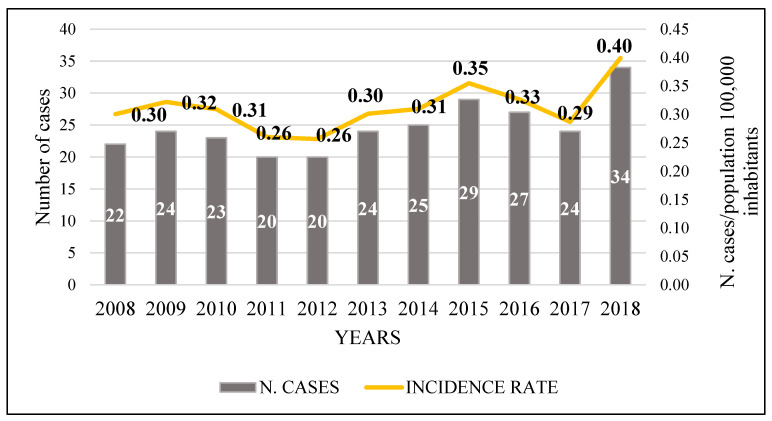
Distribution of the number of new cases of cryptococcosis per year and the incidence rate by state population per 100,000 inhabitants, 2008–2018, JBBUH, Pará, Brazil.

**Figure 2 microorganisms-10-01411-f002:**
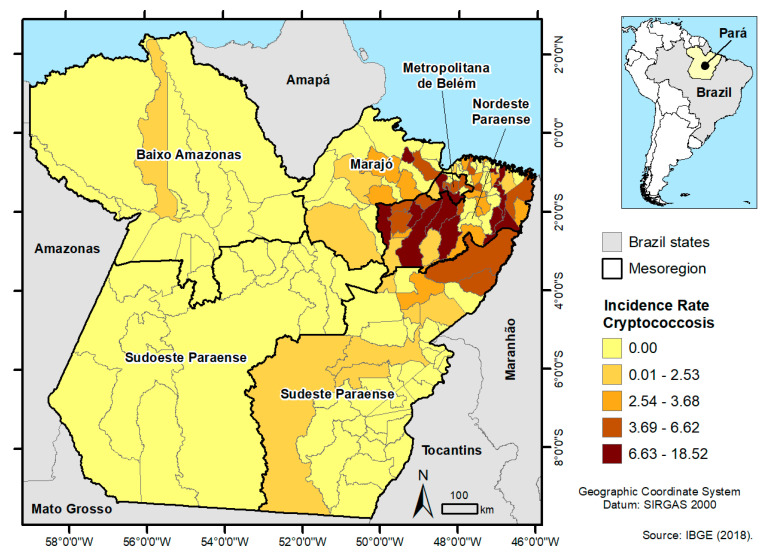
Spatial distribution of the incidence rate of cryptococcosis according to the population of the mesoregions of the Pará per 100,000/inhabitants. (IBGE). 2008–2018, JBBUH, Pará, Brazil.

**Table 1 microorganisms-10-01411-t001:** Sociodemographic profile of patients with cryptococcosis, 2008–2018, JBBUH, Pará, Brazil.

Variables	N	%
Sex		
Female	103	38.0
Male	169	62.0
Age Range		
13–23	53	19.5
24–34	98	36.0
35–45	77	28.3
46–56	33	12.1
57–67	11	4.0
Schooling		
HS incomplete	176	64.7
HS complete	64	23.5
UE incomplete	9	3.3
UE complete	8	2.9
No registry	15	5.5
Professional occupations		
Household activities	54	22.0
Agricultural activities	34	13.8
Student	32	13.0
Self-employed	15	6.1
Merchant	13	5.3
Bricklayer	11	4.5
Teacher	11	4.5
Unemployed	9	3.7
Clerk	8	3.3
Driver	8	3.3
Servant	7	2.8
Hairdresser	6	2.4
Others	28	15.4

Source: Authors’ research (2019). HS-high school, UE-university education.

**Table 2 microorganisms-10-01411-t002:** Clinical and laboratory profile of patients with cryptococcosis, 2008–2018, JBBUH, Pará, Brazil.

Variables	N (272)	%
Hospitalization time		
1–15	72	26.5
16–30	53	19.5
31–60	88	32.3
>60 days	59	21.7
Time of disease onset		
Acute < 7 days	80	31
Subacute 7–29 days	120	46
Chronic > 30 days	59	23
Signs and Symptoms		
Headache	245	90.1
Fever	207	76.1
Vomiting	194	71.3
Stiffness of nape	89	32.7
Other symptoms	70	25.7
Laboratory tests		
Direct detection of fungus in CSF	267	98.2%
Cultures	193	70.9%
Tomography	29	10.7%
Biopsy	4	1.5%
CSF analysis		
Cellularity mm^3^		
0–10 cells	70	27.8%
11–500 cells	163	64.7%
501–1000 cells	11	4.4%
>1000 cells	8	3.1%
Predominant cell		
Mononuclear cells	225	88.9%
Polymorphonuclear cells	28	11.1%
Glycorrhachia		
<40	141	55.1%
≥40	115	44.9%
Proteinorrachia		
<40	51	21.4%
40–100	116	48.7%
101–200	47	19.7%
>200	24	10.2%

Source: Authors’ research (2019).

**Table 3 microorganisms-10-01411-t003:** Antifungal treatment applied to patients with cryptococcosis according to clinical evolution, 2008–2018, JBBUH, Pará, Brazil.

Antifungal Treatment	Clinically Improved N = 145	%	Death N = 70	%	Total	*p*-Value
AmB/Fluconazole	115	78.7%	31	21.3%	146	<0.0001 ^a^
Fluconazole	5	83.3%	1	16.7%	6	
AmB	25	39.7%	38	60.3%	63	

Source: Authors’ research (2019). Amphotericin B deoxycholate (AmB). ^a^ Test G.

**Table 4 microorganisms-10-01411-t004:** Clinical evolution of patients with cryptococcosis according to the present immunological condition, 2008–2018, JBBUH, Pará, Brazil.

Clinical Evolution	Hiv− N = 102	%	Hiv+ N = 170	%	*p* Value
Clinically improved	73	71.6%	90	52.9%	0.0036 ^b^
Mortality	29	28.4%	80	47.1%	

Source: Authors’ research 206 (2019). ^b^ Chi-square test.

## Data Availability

The database used and/or analysed during the current study is not publicly accessible but can be available, upon reasonable request, from the corresponding authors.

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
