# Peer review of "Cryptococcosis: Identification of Risk Areas in the Brazilian Amazon"

_microorganisms, 2022, doi:10.3390/microorganisms10071411_

Round 1

Reviewer 1 Report

This is a report about 272 patients with cryptococcal meninigitis seen at a hospital in Belem, Brazil between  2008 and  2018. We are told the patients'  age, sex, occupation, education, mortality, site of infection,  year of admission, signs and symptoms, length of hospital stay, CSF lab values  and HIV status. We are not told their underlying disease other than HIV or the cryptococcal species (C. neoformans or gattii  species complex).

A significant issue lies in the interpretation of the data. The higher incidence in young men living in the urban areas should have  been correlated with the population density of HIV-positive patients, which comprised  170 of the  272 patients, not the incidence in the general population. Lines 214-215 give the incidence of HIV in Belem a 23.6 per  100,000 inhabitants. The authors calculate that their 272 cases would be 3.4 cases of cryptococcosis per 100,000.  Can the authors use data like this to calculate the incidence of cryptococcosis in the HIV patients in their area? Another issue is the use of unclear terms, which could be a translation problem. What is diagnosis by “tomography” (table 2) or disease manifestation as “fungal” (  line 128)? How is “neurological form” different from “disseminated” (line 24 and 127-128)? What is meant by “loss of lung function” (line 131)? Why is  the use of amphotericin B said to be “replaced by more modern alternatives”, considering that amphotericin B is the most potent drug we have (line199)? What does this phrase mean “linked to problems with the means of transporting goods” (line 252)?  The  proposal that the data in the paper could be used to “sensitize competent bodies” (line 294) needs more detail.

The Discussion is far too long. The paper has numerous details that are not related to the topic but general information about the region, such as deforestation. The tables are listed as “supplementary material” (line 299), which is apparently incorrect.

Author Response

Response to Reviewer 1 Comments

Point 1: We are not told their underlying disease other than HIV or the cryptococcal species (C. neoformans or gattii  species complex).

Response 1:

In the research, hospitalized patients diagnosed with cryptococcosis were considered as inclusion criteria. HIV coinfection was cited due to the high number of cases.

Regarding the identification of the species, it was not possible to make this isolation due to the great diagnostic laboratory limitation in the institution.

Point 2: The higher incidence in young men living in the urban areas should have  been correlated with the population density of HIV-positive patients, which comprised 170 of the 272 patients, not the incidence in the general population.

Lines 214-215 give the incidence of HIV in Belem a 23.6 per 100,000 inhabitants. The authors calculate that their 272 cases would be 3.4 cases of cryptococcosis per 100,000. 

Can the authors use data like this to calculate the incidence of cryptococcosis in the HIV patients in their area?

Response 2:

The research objective was to identify the areas of risk for cryptococcosis in the state of Pará.

The incidence rate was calculated as the number of patients with cryptococcosis divided by the population of the municipalities of origin, multiplied by 100,000 inhabitants.

Point 3: What is diagnosis by “tomography” (table 2)?

Response 3: Adjusted description in text to laboratory tests.

10.7% showed the formation of cryptococcoma, both in pulmonary and neurological involvement through tomography.

Point 4: How is “neurological form” different from “disseminated” (line 24 and 127-128) or disease manifestation as “fungal” ( line 128)?

Response 4: Adjusted description in text.

Meningoencephalitis was the most common form of cryptococcosis infection (89,7%), followed by cases associated with pulmonary form (6.6%), the third most frequent was deep fungal infection causing fungemia (1.8%) and cutaneous forms (1.1%).

Point 5: What is meant by “loss of lung function” (line 131)?

Response 5: Adjusted description in text.

Regarding the disease sequelae, 30 (18.4%) cases reported visual deficits, followed by hearing deficits (4.3%), limb paresis and paralysis (3.7%), and pulmonary sequelae (2.5%) (pulmonary cryptococcoma, pulmonary fibrosis and lobectomy).

Point 6: Why is the use of amphotericin B said to be “replaced by more modern alternatives”, considering that amphotericin B is the most potent drug we have (line199)?

Response 6: Adjusted description in text.

AmB lipid formulations have been associated with a lower risk of nephrotoxicity; however, there is still controversy about the differences between the two available lipids formulations.

Point 7: What does this phrase mean “linked to problems with the means of transporting goods” (line 252)? 

Response 7: Adjusted in text.

In addition, the construction of highways allowed the migratory flow from the coast to the interior to the detriment of the exploitation of forest resources.

Point 8: The proposal that the data in the paper could be used to “sensitize competent bodies” (line 294) needs more detail.

Response 8: Adjusted in text.

Thus, the present study aimed to alert the competent bodies (surveillance services, health departments, government agencies) for the need to include cryptococcosis in the National Compulsory Notification List, despite being a regional estimate.

Point 9: The Discussion is far too long. The paper has numerous details that are not related to the topic but general information about the region, such as deforestation.

Response 9: The discussion contests the results found with other authors, regarding the clinical and epidemiological profile of the disease in the Amazon.

In addition, it compares data from hiv-aids patients with other regions of Brazil. And it presents characteristics of the risk areas of its territorial occupation with deforestation.

Point 10: The tables are listed as “supplementary material” (line 299), which is apparently incorrect.

Response 10: excluded.

Reviewer 2 Report

Dear Authors,

in my opinion your work is very interesting in a practical and cognitive context and contributes a lot to medical mycology, pharmacology and especially epidemiology of cryptococcosis. Moreover, Authors also refer to the prevention of this invasive fungal infection, proposing specific practical solutions, including the need to perform screening tests in HIV positive patients, which are asymptomatic in the case of cryptococcosis. The research conducted by you will certainly draw the attention of the scientific community to the fact that the problem of cryptococcosis is alarming not only in sub-Saharan Africa.

All the tables and figures are appropriate for this type of article. In general, the paper has a logical flow. The abstract well correspond with the main aspects of the work. Nevertheless, as a reviewer I am obligated to pay attention even to less important weak points of this work and all mentioned below comments should be carefully considered.

Lines 15-16

I would like to recommend the sentence correction that is needed. In my opinion, areas of risk have just been identified in the study described within this manuscrip, therefore the proposed sentence below sounds better.

,,In this study were identified the areas of risk for cryptococcosis …”

Line 23

In my opinion should be ,,The mean hospitalization time was …”

Line 24

The mortality rate among patients with cryptococcosis was up to 40%.

Line 26

Dot should be at the end of a sentence.

Line 27

Among keywords I would also see "epidemiological profiles".

Line 55

The authors write that cryptococcosis is the second most common cause of mortality in patients with AIDS, and that cryptococcosis is surpassed only by tuberculosis. Based on publication by Park et al. (2009) this fungal infection is more common than tuberculosis. Maybe Authors meant malaria or pneumocystosis?

Park, Benjamin Ja; Wannemuehler, Kathleen Ab; Marston, Barbara Jc; Govender, Neleshd; Pappas, Peter Ge; Chiller, Tom Ma Estimation of the current global burden of cryptococcal meningitis among persons living with HIV/AIDS, AIDS: February 20, 2009 - Volume 23 - Issue 4 - p 525-530 doi: 10.1097/QAD.0b013e328322ffac

Line 56

In case of reference no. 9 there is a mistake in surname of the author of this publication. To the best of my knowledge should be Massimo Cogliati.

Lines 114-115

There are no spaces between lines 114 and 115.

Line 116, 181 and Table 2

Instead of "length of hospital stay", I would suggest "hospitalization time" or "hospitalization period".

Line 119

To the best of my knowledge should be ,,India ink staining" instead of ,,Chinese ink coloration".

Table 2

Instead of ,,Direct research of fungus in CSF” I would like to recommend ,,Direct detection of fungus in CSF”. Moreover, as I know should be ,,predominant cells”

Lines 141-142

There are no spaces between lines 141 and 142.

Line 191

In my opinion ,,…similar to this study ….” sounds better

Line 217

In my opinion there should be ,,…infections caused by yeast-like fungi of the Cryptococcus species complex” not ,,mycoses”

Line 261

,,thousand km2” and ,,inhabitants per km2” should be written with superscript

Author Response

Response to Reviewer 2 Comments

Point 1: Lines 15-16

I would like to recommend the sentence correction that is needed. In my opinion, areas of risk have just been identified in the study described within this manuscrip, therefore the proposed sentence below sounds better.

,,In this study were identified the areas of risk for cryptococcosis …”

Response 1: Adjusted

Point 2: Line 23

In my opinion should be ,,The mean hospitalization time was …”

Response 2: Adjusted

Point 3: Line 24

The mortality rate among patients with cryptococcosis was up to 40%.

Response 3: Adjusted

Point 4: Line 26

Dot should be at the end of a sentence.

Response 4: Adjusted

Point 5: Line 27

Among keywords I would also see "epidemiological profiles".

Response 5: Adjusted

Point 6: Line 55

The authors write that cryptococcosis is the second most common cause of mortality in patients with AIDS, and that cryptococcosis is surpassed only by tuberculosis. Based on publication by Park et al. (2009) this fungal infection is more common than tuberculosis. Maybe Authors meant malaria or pneumocystosis?

Park, Benjamin Ja; Wannemuehler, Kathleen Ab; Marston, Barbara Jc; Govender, Neleshd; Pappas, Peter Ge; Chiller, Tom Ma Estimation of the current global burden of cryptococcal meningitis among persons living with HIV/AIDS, AIDS: February 20, 2009 - Volume 23 - Issue 4 - p 525-530 doi: 10.1097/QAD.0b013e328322ffac

Response 6: Not changed

Text references a more up-to-date manuscript by Park, 2017.

Rajasingham R, Smith RM, Park BJ, Jarvis JN, Govender NP, Chiller TM, et al. Global burden of disease of HIV-associated cryptococcal meningitis: an updated analysis. The Lancet Infectious Diseases, London, Aug. 2017. 17 (8): 30243 – 30248. Available at: https://www.ncbi.nlm.nih.gov/pubmed/28483415

Point 7: Line 56

In case of reference no. 9 there is a mistake in surname of the author of this publication. To the best of my knowledge should be Massimo Cogliati.

Response 7: Adjusted

Point 8: Lines 114-115

There are no spaces between lines 114 and 115.

Response 8: Adjusted

Point 9: Line 116, 181 and Table 2

Instead of "length of hospital stay", I would suggest "hospitalization time" or "hospitalization period".

Response 9: Adjusted to hospitalization time

Point 10: Line 119

To the best of my knowledge should be ,,India ink staining" instead of ,,Chinese ink coloration".

Response 10: Adjusted

Point 11: Table 2

Instead of ,,Direct research of fungus in CSF” I would like to recommend ,,Direct detection of fungus in CSF”. Moreover, as I know should be ,,predominant cells”

Response 11: Adjusted

Point 12: Lines 141-142

There are no spaces between lines 141 and 142.

Response 12: Adjusted

Point 13: Line 191

In my opinion ,,…similar to this study ….” sounds better

Response 13: Adjusted

Point 14: Line 217

In my opinion there should be ,,…infections caused by yeast-like fungi of the Cryptococcus species complex” not ,,mycoses”

Response 14: Adjusted

Point 15: Line 261

,,thousand km2” and ,,inhabitants per km2” should be written with superscript

Response 15: Adjusted
